# Synaptic Dysfunction and Plasticity in Amyotrophic Lateral Sclerosis

**DOI:** 10.3390/ijms24054613

**Published:** 2023-02-27

**Authors:** Rosario Gulino

**Affiliations:** Department of Biomedical and Biotechnological Sciences, Physiology Section, University of Catania, 95123 Catania, Italy; rosario.gulino@unict.it

**Keywords:** ALS pathogenesis, compensatory mechanisms, disease onset, homeostatic synaptic plasticity, motoneuron disease, structural plasticity

## Abstract

Recent evidence has supported the hypothesis that amyotrophic lateral sclerosis (ALS) is a multi-step disease, as the onset of symptoms occurs after sequential exposure to a defined number of risk factors. Despite the lack of precise identification of these disease determinants, it is known that genetic mutations may contribute to one or more of the steps leading to ALS onset, the remaining being linked to environmental factors and lifestyle. It also appears evident that compensatory plastic changes taking place at all levels of the nervous system during ALS etiopathogenesis may likely counteract the functional effects of neurodegeneration and affect the timing of disease onset and progression. Functional and structural events of synaptic plasticity probably represent the main mechanisms underlying this adaptive capability, causing a significant, although partial and transient, resiliency of the nervous system affected by a neurodegenerative disease. On the other hand, the failure of synaptic functions and plasticity may be part of the pathological process. The aim of this review was to summarize what it is known today about the controversial involvement of synapses in ALS etiopathogenesis, and an analysis of the literature, although not exhaustive, confirmed that synaptic dysfunction is an early pathogenetic process in ALS. Moreover, it appears that adequate modulation of structural and functional synaptic plasticity may likely support function sparing and delay disease progression.

## 1. Introduction

Severe loss of spinal and bulbar motoneurons (MNs) is the main feature of different MN diseases, including amyotrophic lateral sclerosis (ALS) and spinal muscular atrophy. ALS is a rare but severe and fatal disease with an incidence of approximately 2 people per 100,000 [1]. The majority of ALS patients have spinal onset characterized by focal muscle weakness. Bulbar onset occurs in some patients showing dysarthria or dysphagia. Paralysis is progressive and leads to death due to respiratory failure within 2–5 years after diagnosis [2]. Although motor dysfunction represents the primary symptom of ALS, up to 50% of patients also develop cognitive deficits, and approximately 10–15% of patients are affected by a complex syndrome including motor, cognitive, and behavioral symptoms, known as ALS with frontotemporal dementia (ALS/FTD) [3,4,5].

Despite considerable efforts in ALS research, the causes of the disease are still unknown and no effective cure is available, but many recent data are challenging the vision of a simple MN disease, suggesting that ALS could be caused by a network failure [6,7].

Although MN loss is the critical hallmark of ALS, given the specific vulnerability of these cells, other cell populations are involved, including corticospinal neurons as well as glial and muscle cells [1,8,9,10], but their crosstalk remains unclear. Glutamate excitotoxicity, oxidative stress, and the role of different genetic alterations, including mutations of the genes encoding for Cu/Zn superoxide dismutase 1 (SOD1), TAR-DNA binding protein of 43 kDa (TDP-43), fused in sarcoma (FUS), and C9orf72, have been documented in animal models and in patients with both familial and sporadic ALS [9,10]; however, the interplay between these hallmarks is largely unknown. Among these, the C9orf72 mutation is the most frequent genetic ALS hallmark, and TDP-43 dysfunctions and aggregates are common pathological features in the majority of both familial and sporadic ALS [11]. A number of transgenic animal models have been developed in order to dissect the pathological mechanisms and test novel therapeutic approaches [12,13].

By modeling ALS incidence data, recent studies by Al-Chalabi and colleagues have proposed that ALS is a six-step process whereby the onset of symptoms occurs only after being exposed to a sequential series of risk factors [14,15] and genetic mutations, when present, account for 1–4 of these steps [15]. A recent study has shown that this multi-step process is also valid for the SOD1 transgenic mouse model [16]. Indeed, approximately 10% of ALS cases are familial (fALS), with a clear genetic origin and inheritance, whereas the vast majority of ALS cases are apparently sporadic (sALS), although mutations in ALS-linked genes can be found in all ALS cases and a clear-cut boundary between the two ALS forms does not exist [1,2,6,7,17]. Large heterogeneity characterizes the clinical manifestations of ALS, in terms of age, site of onset, progression speed, and comorbidity, and this may depend on different pathogenic mechanisms driven by different genetic backgrounds and environmental conditions [18]. It seems probable that ALS pathogenesis could have a focal origin in a region of the central nervous system in which multiple factors contribute to the creation of a toxic milieu [1,9,19]. After onset, the progression of the disease could be driven, at least in part, by a prion-like propagation of misfolded proteins, including TDP-43 [10,20,21], but many other pathological processes are involved.

It is known, for instance, that an important role during pathogenesis could be exerted by plasticity occurring within neuronal circuitries. Similar to other well-known homeostatic systems, including those regulating blood pressure, glucose metabolism, or temperature, there are a variety of mechanisms regulating the activity of neural networks within an adequate range. One of these mechanisms is known as homeostatic synaptic plasticity, which serves as a modulator of synaptic strength in order to prevent excessive potentiation or depression of synapses within a given neuronal network, thus maintaining the efficiency and fidelity of signaling and avoiding cell suffering [22,23,24,25,26]. Several other homeostatic mechanisms are also responsible for the stabilization of neuronal circuitries, including balance between excitatory and inhibitory inputs [27], modulation of neuronal excitability [28,29] and structural re-shaping of circuitries by regulating the number, size, and distribution of synaptic connections [26,30].

These mechanisms may be responsible for adaptive responses that compensate for the progressive loss of function caused by neurodegeneration, thus delaying symptom onset [31,32,33,34,35,36,37]. Therefore, symptoms would likely appear only when these adaptive mechanisms become unable to balance the loss of a high number of neurons. Thus, it may be argued that a sudden collapse of compensatory processes represents one of the unknown steps towards disease, predicted by the mathematical model proposed by Al-Chalabi and colleagues [14,15]. Plasticity can likely also take place after disease onset and, although unable to hide the functional loss, it may contribute to increased lifespan. On the other hand, the disruption of these homeostatic processes could be responsible for neuronal suffering and other pathological events triggering neurodegeneration [38].

This comprehensive review is an overview of the experimental findings exploring the controversial role of synaptic plasticity and its failure in ALS, employing either human patients and autoptic samples or animal models of ALS. Despite the limitations of animal models, which only partially recapitulate the human disease, these studies have the advantage of better dissecting pathological events at the cellular and molecular levels and, importantly, facilitating observation of the pathological processes occurring during the presymptomatic stage. The events occurring at early stages during pathogenesis are extremely important because they demonstrate the resiliency of neural circuits, and they are probably the most important processes to be studied in order to dissect the nature of compensatory plastic changes during disease.

## 2. Pathogenic Aspects of Synaptic Plasticity vs. Its Adaptive Role in Resiliency of Neural Circuits

Structural and/or functional modifications of specific synapses or large neuronal networks have been documented as disease-linked mechanisms either in patients or in different experimental models of ALS. These pathological changes of synaptic connectivity were found at all levels of the voluntary motor system, from the sensorimotor cerebral cortex down to the neuromuscular junction (NMJ) and skeletal muscle (Table 1). Despite the involvement of these mechanisms in ALS pathogenesis, many studies have suggested that the same processes could represent compensatory events capable of counteracting disease progression, thus delaying onset and mitigating the functional effects of neurodegeneration, but the boundary between beneficial and detrimental effects of synaptic modifications is extremely confused.

### 2.1. Cerebral Cortex

Severe functional impairment and cell death take place in the cerebral cortexes of patients affected by ALS-FTD, but also in ALS patients with or without cognitive or behavioral symptoms [3,4,5,93]. Connectome studies using functional magnetic resonance imaging (fMRI) on ALS patients have found a decrease in connectivity among different cortical areas (frontal, parietal, temporal, and occipital cortex), which was associated with disease progression [39]. Conversely, another fMRI study found increased functional connectivity between the sensorimotor cortex and other brain areas that was reversed in the advanced stage of disease [42]. Together, these studies suggested that a severe alteration of network topology may occur in the whole cerebral cortex during ALS pathogenesis and progression, and that some of these changes could represent a compensatory, although transient, adaptation of cortical circuits. This altered connectivity may be accompanied by hyperactivation of different cortical areas. For instance, the reduction of hand grip strength during ALS progression is paralleled by overactivation of the somatosensory cortex, which decreases with disease progression [43], and this could be interpreted both as a marker of disease progression and as a compensatory process that possibly delays, although weakly, functional deterioration. Despite the lack of information about possible presymptomatic modifications in ALS patients, other authors have reported hyperactivation in frontal areas of ALS patients with cognitive dysfunction [44] or in the primary, premotor, and supplementary motor cortex and subcortical structures (cerebellum, basal ganglia, and brainstem) during both the execution of motor tasks and its imagery [45,46,47]. These cortical plastic changes were interpreted as compensatory events trying to sustain functions, but failing, as the neurodegenerative process passed a critical threshold.

Alterations of connectivity and synaptic function in the brain have also been reported in several animal models recapitulating different pathological aspects of ALS, and these studies have also proposed different cellular and molecular mechanisms. An increase of intrinsic cortical excitability has been reported in mice overexpressing an SOD1 mutation, but the pattern and timing of the cell-type specific alterations appeared to be intricate [53]. In particular, increased excitability was observed in different neurons in the motor cortex, including corticospinal neurons and inhibitory interneurons, which emerged at the early presymptomatic stage but were partially normalized during disease progression and re-emerged in the later stage of disease, thus suggesting an attempt to normalize neuronal activity, although transient. Moreover, these events were paralleled by differential gene expression in different cell types, which may partially explain the selective vulnerability to neurodegeneration [53]. Excitability of cortical neurons is linked to glutamatergic signaling through both N-methyl-D-aspartate (NMDA) and alpha-amino-3-hydroxy-5-methyl-4-isoxazole propionate (AMPA) receptors. Networks of in vitro cultured glutamatergic cortical neurons, obtained from patient-derived induced pluripotent stem cells with the C9orf72 mutation, have shown increased neuronal activity associated with higher synapsin-I expression but lower burst duration, attributable to a reduced number of readily releasable vesicles and impaired AMPA-mediated synaptic plasticity [40], suggesting that early alterations of synaptic function may precede cortical neurodegeneration in patients. However, this finding appears in contrast with other data showing that riluzole can attenuate excitotoxicity in cortical neurons by reducing the readily releasable vesicle pool [41]. Moreover, a decrease in the expression of NMDA was found in the motor cortex of presymptomatic SOD1 transgenic mice together with a reduction in dendritic arborization and defects in long-term potentiation (LTP) [48].

Many other studies have proposed various molecular mechanisms of hyperexcitability and cortical dysfunction using different ALS mouse models. For instance, dysregulation of FUS protein and its cytoplasmic accumulation are signs of ALS and ALS/FTD, and mutations of the gene can cause the disease [94,95]. Normally, this protein is primarily nuclear and widely expressed in the central nervous system (CNS), but it is also present in presynaptic terminals in close proximity to synaptic vesicles [49]. FUS mutant mice showed accumulation of FUS protein in the presynaptic compartment of synaptic contacts to dendrites and the bodies of cortical and hippocampal neurons, accompanied by alterations of mRNAs regulating synaptic functions, impaired GABAergic synapses, and NMDA signaling that partially explained the hyperexcitability and neurodegeneration [52]. These results were confirmed in cholinergic brain neurons from FUS mutant Drosophila, showing a reduced number of synaptic mitochondria, decreased synaptotagmin levels, loss of neuronal projections, impaired axonal transport, and neuronal hyperexcitability [96].

TDP-43 proteinopathy is present in the majority of ALS cases [11]. Like FUS, this mRNA-binding protein is normally present in the nucleus, but it has recently also been found in the post-synaptic compartment of dendritic spines in cortical neurons where it forms ribonucleoprotein (RNP) granules that block local translation. Neuronal activity unbinds TDP-43 and permits the translation and regulation of synaptic function [54]. These activity-dependent RNP granule dynamics are hampered in mice overexpressing TDP-43, thus causing impaired synaptic function and plasticity as well as TDP-43 accumulation and symptoms of ALS/FTD [54]. Similar results were obtained using a conditional knockout mouse with TDP-43 depletion in the forebrain. In particular, this mouse model showed alterations of synaptic plasticity, inflammation, reduced LTP in the hippocampus, cell death, and symptoms of ALS/FTD, thus demonstrating that these deficits may depend not only on the toxicity of mislocalized TDP-43 but also on the protein’s loss of function [58]. Dendritic spine turnover and morphological changes are events of synaptic plasticity that take place during the normal functioning of neurons. Mice overexpressing a mutant form of TDP-43 showed alterations of dendritic spine plasticity specifically in the motor cortex, but not in the somatosensory cortex [55]. Interestingly, these alterations were less severe in female mice, together with better disease outcome, and these effects were linked to estrogen levels [55].

Taken together, this collection of studies demonstrates the complex involvement of structural and functional plastic changes within cortical networks, which can either represent a piece of the pathogenic process, a compensatory mechanism demonstrating the functional resiliency of neural circuits, or both. Considering the central role of corticospinal neurons in the context of ALS pathogenesis, further research efforts are required to clarify the link between synaptic functions and the selective vulnerability of these neurons.

### 2.2. Hippocampus

The hippocampus and its connections with other brain areas are fundamental for the normal functioning of learning and memory processes, as well as for mood and other cognitive functions. Together with the above-described cortical modifications, functional deficits, cell death and plastic changes are also present in the hippocampal circuitry in different neurodegenerative disorders characterized by cognitive dysfunctions, including Alzheimer’s disease [97,98,99,100,101], ALS, and FTD [102,103]. Few studies have addressed hippocampal involvement in ALS and FTD patients. However, autoptic examination of brains of individuals with ALS/FTD caused by the C9orf72 mutation demonstrated hippocampal sclerosis in a number of cases [102] and TDP-43 proteinopathy in many CNS areas, including the neocortex, lower motor neurons, and hippocampus [103].

Neurogenesis is present in adult mammals but restricted to a few CNS areas, including the hippocampus. Adult hippocampal neurogenesis is an important form of plasticity occurring in the dentate gyrus, which is linked to learning and memory [104] and a variety of other cognitive functions in humans [105,106]. Alterations in adult hippocampal neurogenesis have been linked with aging and neurodegenerative diseases [104], including ALS and FTD [59]. Autoptic analysis of human brains has revealed morphological changes in the dentate gyrus, with altered proliferation, differentiation, vascularization, and gliosis, and these neurogenerative processes may underlie cognitive defects in ALS and FTD [59]. A reduction in hippocampal neurogenesis, accompanied by decreased hippocampal LTP and LTD, was also observed in C9orf72 knockout mice [60]. Additionally, a marked reduction in synapsin levels was found in the hippocampus of ALS/FTD patients with the C9orf72 mutation, and evidence from in vitro and in vivo studies on C9orf72 mutant mice suggested an interaction between C9orf72 and synapsin, with a consequent reduction in synapsin levels at presynaptic terminals and decreased synaptic density in the hippocampus [61]. Synaptic localization of C9orf72 transcripts is also important for the regulation of glutamate receptor levels in the hippocampus, thus likely contributing to both synaptic function and excitotoxicity [62].

Many other studies have used different animal models to dissect the role of hippocampal synaptic function and plasticity in ALS, applying a deeper mechanistic approach and obtaining more cellular and molecular details. For instance, alterations of synaptic functions in the hippocampus and neocortex have been reported in mice carrying an FUS mutation as a result of dysregulation of RNA trafficking and alterations of protein homeostasis, mitochondrial function, and axonal protein synthesis, thus causing dysfunctions in brain connectivity and cognition [49,50,51,52]. Interestingly, alterations of RNP granule assembly were found in Drosophila with the ataxin-2 mutation together with changes in long-term memory; this gene can also modify the ALS phenotype in C9orf72- and FUS-mutant Drosophila models [63].

ALS and FTD are also linked to TDP-43 proteinopathy, which is present in both familial and sporadic cases, where cytoplasmic aggregates of abnormally phosphorylated and ubiquitinated TDP-43 are present in neurons and glial cells of various CNS areas, thus causing neurodegeneration due to both toxicity and loss of function of mutant TDP-43 [107]. In fact, mice overexpressing TDP-43 [56,57] or conditional knockout mice with depletion of TDP-43 in the forebrain [58] both showed dementia-like behavioral changes together with reduced synaptic density and LTP in the hippocampus (as well as in the cortex) and hippocampal atrophy. The mechanisms underlying the effects of TDP-43 proteinopathy on synaptic functions and plasticity are intricate and further research efforts are necessary. Many other factors interact with TDP-43, including, for example, the molecular chaperone Cyclophilin A and the mitochondrial protein CHCHD10 (coiled-coil-helix-coiled-coil-helix domain containing 10), and their alterations can drive TDP-43 proteinopathy and synaptic defects similar to those caused by alterations of TDP-43 itself, including a reduction in synaptic proteins and hippocampal LTP together with motor unit loss [56,64]. In contrast with the above-described effects, conditional knockout rats with TDP-43 depletion in neurons showed an improvement in fear conditioning, thereby revealing an increase in memory formation. These effects were associated with the extended duration of hippocampal LTP, which conversely may depend on the observed enlargement of dendritic spines in hippocampal CA1 neurons and rearrangement of AMPA receptors [65]. The question of whether this increased synaptic plasticity could be considered as a pathogenic process, a compensatory adaptation counteracting other defects in neuronal activity, or both, is an interesting matter of debate and future studies.

Similar considerations may arise from other studies showing an increase of hippocampal LTP in presymptomatic SOD1 transgenic mice [66,67,68], which could be either considered as compensatory changes or as pathogenic processes leading to degeneration due to excitotoxicity [66]. Synaptic and cognitive changes were associated with higher expression of hippocampal GluR1 AMPA receptor subunits [66] and degeneration of parvalbumin-positive neurons [67], which were followed by a decrease in LTP and levels of NMDA receptors in the symptomatic stage [68].

### 2.3. Brainstem and Spinal Cord

As described above, ALS is a complex neurodegenerative disease characterized by both motor and non-motor dysfunctions, and functional impairments and neurodegeneration can be present at all levels of the central and peripheral nervous systems. However, the onset of disease is usually defined by focal neuromuscular deficits, with muscle weakness and atrophy accompanied by the death of defined subpopulations of spinal or brainstem MNs. Thus, the onset may be spinal, with muscle weakness affecting arms or legs, or bulbar, with early symptoms of dysarthria or dysphagia [1,2]. Therefore, lower MNs are among the most vulnerable cell types and the linkage between synaptic functioning and MN degeneration is of crucial importance. Several studies have reported events of synaptic alteration and plasticity in different phases of ALS pathogenesis. The assessment of H-reflex in ALS patients has suggested that both recurrent inhibition of MNs (by Renshaw cells) and post-activation depression (by inhibitory interneurons controlling primary afferents from spindles) may be reduced during disease, thus likely contributing to MN overexcitability and excitotoxicity [69]. Autoptic examination of the spinal cord of patients with ALS revealed a reduction of MN body area and the number of synaptic contacts to the MN body and dendrites, but, interestingly, synaptic size was not diminished, thus suggesting a compensatory adaptation that preserves motor function during the early stage of denervation [70]. Similarly, a significant age-dependent reduction of dendrites and synaptophysin was found in the anterior horn of ALS patients, with signs of reinnervation during disease progression and dramatic synaptic loss in the late stage of disease [71].

Studies involving animal models have further investigated these initial findings obtained from patients. For instance, disease-linked changes in MN soma size were also observed in SOD1 transgenic mice. In particular, an enlargement of MN soma size has been observed as early as the presymptomatic stage of disease, followed by a dramatic size reduction below control levels at the end stage [31,33], which confirmed the findings in symptomatic patients. Given that the increase in soma size was higher for fast-fatigable MNs, which are also the most vulnerable to degeneration [31], the question of whether this can be considered as disease-linked or a compensatory plastic process remains open. However, the increase in soma size was accompanied by the sprouting of proximal cholinergic processes that also collapsed at the end stage [33] and was linked to MN excitability and metabolism. Therefore, although detrimental effects cannot be excluded given the observed reversal of these mechanisms at the end stage, it appears likely that these early modifications can be plastic changes that partially and transiently compensate for MN loss and pathological changes in MN excitability [31,33]. On the other hand, it has been observed that increased MN excitability can be induced by altered glutamate signaling, which may trigger excitotoxicity and neurodegeneration, and that these effects can be rescued by riluzole [72].

Clear evidence of compensatory synaptic plasticity has been reported in the spinal cord of SOD1 transgenic mice, where cholinergic C-type synaptic terminals to the soma and proximal dendrites of MNs were preserved and increased in size, whereas the number of S- and F-type boutons progressively decreased together with MN degeneration, starting at the presymptomatic stage [73,74]. C-bouton enlargement was also found in autoptic spinal cord tissue from ALS patients [74]. Differently, although not necessarily in contrast to these observations, another study reported that C-boutons were lost in both SOD1 mice and ALS patients and that viral-mediated delivery of Neuregulin-1 to SOD1 mice could restore C-boutons and increase survival [75]. Moreover, elongation and overbranching of MN dendrites were found in SOD1 transgenic mouse spinal cords starting at the perinatal stage, which was also interpreted as a consequence of early alterations of synaptic inputs and MN excitability [76]. Indeed, early synaptic dysfunctions were present in SOD1 mouse spinal cords, as a consequence of impaired oxidative metabolism in the presynaptic compartment [108]. On the other hand, mice carrying a mutation of the gene encoding vascular endothelial growth factor (VEGF) developed an ALS-like phenotype, and transcriptomic analysis showed a massive downregulation of numerous genes linked to crucial mechanisms, including synaptic functions, axonogenesis, axonal transport, and growth factor signaling pathways, thus impairing plastic changes in a pathogenic context requiring plasticity and triggering neurodegeneration [77].

Again, structural and functional synaptic modifications can represent either a piece of the pathogenic mechanism or a trait of compensatory capability (or both). In any case, this capability could be considered as a putative therapeutic target under certain conditions.

The ability of surviving MNs to compensate for the loss of an MN subpopulation and maintain motor function has also been demonstrated in a mouse neurotoxic model of MN loss. In this model, the partial removal of lumbar MNs caused an acute functional impairment that was followed by recovery, which conversely may have been linked to events of synaptic plasticity (as suggested by changes in the expression of synapsin-I, synaptophysin, and AMPA receptor subunits, as well as by electromyography) occurring within the spinal circuitries and regulated by several factors, including TDP-43, sonic hedgehog, and other factors known to also regulate neural development and stem cell functioning [34,35,36,37,79,80,81]. Another interesting example of compensatory synaptic plasticity is phrenic long-term facilitation that preserves respiratory function despite the progressive death of phrenic MNs, but it fails at the end stage when respiratory failure is imminent. Recent findings suggested that this mechanism can be linked to synaptic modifications of serotonin receptors from the presymptomatic to end stages in SOD1 transgenic rats and can be induced by acute intermittent hypoxia [82,83].

ALS patients frequently develop dysphagia and dysarthria as a consequence of bulbar MN death [1,2], whereas eye movements are generally spared because of different vulnerabilities of these subpopulations of brainstem MNs [109]. Similar to that described for spinal MNs, plasticity is also present in central and peripheral synapses of bulbar MNs [109,110,111,112]. Interestingly, although neurogenesis is normally rare in the brainstem, it can be increased in both pre- and post-symptomatic SOD1 transgenic mice [78].

### 2.4. NMJ and Skeletal Muscle

NMJ degeneration is an early event in ALS pathogenesis, starting before the onset of symptoms, and such alterations of the NMJ have been reported in several studies involving different animal models [85,86,113]. The delay of symptom appearance despite early NMJ alterations suggests the occurrence of compensatory adaptations. Indeed, the NMJ is capable of profound structural and functional plasticity under different physiological and pathological conditions [114,115]. It has been also found that slow-type motor units display extensive NMJ compensatory sprouting, whereas fast-fatigable motor units do not show this capability and appear to be more vulnerable to neurodegeneration and subjected to earlier loss [85,87,88,114]. It has been shown in SOD1 transgenic mice that morphological changes of the NMJ are preceded by functional modifications. In particular, higher and lower quantal contents of neurotransmitters have been found in the NMJ of slow and fast-fatigable motor unit types, respectively, at the early presymptomatic stage [88]. These signs persisted at the preonset stage, when anatomical alterations of the NMJ and MN death were already present [88].

The mechanisms controlling NMJ synaptic modifications include the participation of Schwann cells [116,117]. In particular, the selective expression of Semaphorin 3A in Schwann cells associated with fast-fatigable motor units could partially explain the lack of sprouting in their NMJ as well as in the SOD1-G93A mouse model of ALS [114]. Moreover, NMJ-associated Schwann cells were found to have reduced muscarinic activation after injury and this behavior could drive NMJ reinnervation; interestingly, this mechanism of Schwann cell plastic adaptation after injury was absent in SOD1 mice [118].

The temporal sequence of functional and morphological alterations of the neuromuscular systems would suggest a dying-back pattern of neurodegeneration, as NMJ alterations usually appear before MN death [88]. However, this hypothesis is debated. It is known, for instance, that target-derived neurotrophins exert potent synaptotrophic action in the adult CNS and are involved in synaptic plasticity [119,120,121,122,123], while target disconnection may cause synaptic stripping due to the lack of trophic support [124]. Retrograde trophic support can also be crucial in the pathogenesis of neurodegeneration [84,89], and it could represent one of the reasons for the differential vulnerabilities among different MN subpopulations. Moreover, the accumulation of neurotrophin in muscles of ALS patients could suggest an attempt to support motor unit function, although inefficient [84], and the higher expression of VEGF and nerve growth factor receptors in the NMJ of extraocular motor units can partially support the resistance of extraocular MNs to neurodegeneration in ALS [110,112]. On the other hand, other studies supporting the dying-forward hypothesis suggest that synaptic retraction and muscle atrophy are a consequence of MN suffering. For instance, it has been shown that astrocyte-targeted RNA interference against mutated SOD1 protects fast-fatigable motor units and promotes the rescue of NMJ size and its reinnervation, thus also protecting muscles from atrophy. This evidence suggests that NMJ damage and/or plasticity could be a consequence of MN suffering [125]. Consistently, the application of kainic acid to spinal MNs in an in vitro experimental set-up has shown that these cells are vulnerable to proximal somatodendritic, but not axonal, excitotoxin exposure [126]. Moreover, the in vivo spinal application of kainic acid caused NMJ retraction and MN death, thus suggesting a dying-forward mechanism of neuromuscular degeneration caused by spinal excitotoxic insults [125].

Remarkably, a recent study using mouse and Drosophila models of MN loss provided direct evidence of NMJ homeostatic plasticity, which was mediated by the evolutionary conserved presynaptic ENaC channel. In fact, the deletion of the gene encoding ENaC in these models prevented NMJ plasticity and severely increased MN loss and disease progression [90]. Other studies using Drosophila models carrying mutations of ALS-linked genes have demonstrated the involvement of NMJ plasticity. In particular, the loss of function of the Drosophila homologue of TDP-43 caused alterations of MN synaptic terminals, resulting in motor deficits and reduced lifespan, whereas conditional expression of the gene in deleted animals rescued the phenotype by modulating the expression of presynaptic vesicular proteins [91]. Similarly, the C9orf72 mutation caused a reduction in synaptic arborization and active zone number, with a decreased number of both pre- and post-synaptic components but, despite these dramatic events, the synapses retained the ability to express homeostatic synaptic plasticity. However, although this homeostatic process remained dormant or was not sufficient to restore motor function, it was increased by retrograde feedback from the muscle provided by either overexpression of post-synaptic receptors or specific muscle genes [92].

## 3. Discussion

Structural and functional synaptic alterations and plasticity may occur at all levels of the central and peripheral nervous systems and at all stages of ALS disease. Despite evidence showing that certain cell populations are more vulnerable to cell stress and degeneration, it remains unclear which of these cell types will degenerate first. The historical controversy between the dying-back and dying-forward hypotheses of ALS pathogenesis is still open [127,128,129,130,131,132]. Nevertheless, whatever the nature and direction of stressful insults, the synapse is the main regulator of this traffic and it may represent either the primary site of pathology or a site where the degenerative process could be functionally neutralized, thus allowing transient function sparing. Therefore, although ALS should be considered as a syndrome caused by different etiological drivers, rather than a single disease, synaptic function and plasticity, and/or its failure, would represent a central aspect in all pathogenic mechanisms. In fact, modifications of synaptic functioning have been documented both in patients and in all available animal models mimicking the different aspects of ALS.

Large cortical areas are subjected to anatomical and functional alterations in ALS, as both reductions [39] and increases in cortical connectivity and neuronal excitability [42,43,44,45,46,47] have been found in ALS patients, and these changes may either represent markers of disease progression or spontaneous compensatory adaptations to functional deficits. Several studies have confirmed these intricate aspects of ALS pathogenesis using different animal models, including SOD1 transgenic mice [48,53], FUS mutant mice, or Drosophila [49,50,51,52], as well as mice with TDP-43 deletion [58] or overexpression [54,55,56,57]. For instance, compensatory changes involving glutamatergic brain synapses can either support functions and/or make specific neuronal populations more prone to excitotoxic damage [40,41,48,53]. It is therefore likely that certain adaptive events may be stressful to specific neuronal populations while allowing transient function sparing [53]. According to the dying-forward hypothesis, early alterations and hyperexcitability of corticospinal neurons can cause excitotoxic insults to of the spinal MNs, which can conversely induce peripheral dysfunctions at the NMJ and muscle levels [125,126]. Indeed, MN hyperexcitability, excitotoxicity, altered soma size, and synaptic plasticity were also found in the spinal cords of patients [69,70,71] and animal models [72,77]. However, again, similar events have been interpreted as compensatory changes in many studies, although the pathological implication of these changes, as a result of stressful insults to overactivated synapses and neurons, cannot be ruled out [31,33,34,35,36,37,73,74,75,76,78,79,80,81,82,83].

The NMJ is another crucial crossroad for pathological and adaptive processes in ALS. NMJ alterations and synaptic detachment are early events that often precede MN degeneration, and this is the main evidence supporting the dying-back hypothesis [85,86,91,113], also considering the important role of muscle-derived neurotrophins [84,89]. However, even the NMJ has demonstrated a high capability for structural and functional synaptic plasticity, although these mechanisms may be dormant, transient, or limited to specific NMJ subtypes [87,88,90,92], thus suggesting that experimental approaches aimed at enhancing this intrinsic capability would be promising.

Controversial results also have been reported about hippocampal plasticity. In particular, reduced neurogenesis [59,60], impaired synaptic function and plasticity, defective LTP and memory formation [56,57,63,64], hyperexcitability and excitotoxicity [49,50,51,66] have been shown in ALS and FTD patients and in different animal models. In contrast to these results, increased synaptic plasticity, enlarged dendritic spines, and increased LTP were also found in both SOD1 and TDP-43 transgenic mice [65,66,67,68].

The body of evidence presented here, although obviously incomplete, demonstrates that the roles of synaptic function and plasticity are extremely important in the context of ALS pathogenesis and that this research field is promising and currently growing. Importantly, activity-dependent synaptic plasticity has been implicated in the compensatory processes occurring after a traumatic brain or spinal cord insult, and therapeutic strategies aimed at potentiating this spontaneous capability of the nervous system have proven to be beneficial [133,134,135,136,137,138]. Unfortunately, modifications of synaptic function and plasticity taking place in neurodegenerative diseases, including ALS, seem to maintain a double nature of beneficial and detrimental mechanisms. In particular, it appears that structural and functional synaptic modifications occurring in the context of ALS pathogenesis, from the early presymptomatic to advanced stages, can be classified in the following three categories: (i) synaptic deficits disrupting the normal functioning of neural circuits and contributing to disease development and progression; (ii) synaptic plasticity being dormant during pathogenesis, or not sufficient to sustain functions, but potentially activatable; (iii) anatomical and functional synaptic remodeling that is capable of preserving function despite the progressive loss of neurons and connections in the neural circuits. Obviously, even the more efficient plastic changes will only be able to compensate for the effects of cell death until a given threshold, with symptom onset probably occurring when neurodegeneration reaches this threshold.

## 4. Conclusions and Future Directions

From the analysis of the literature presented here, it appears that synaptic function is a central aspect of ALS pathogenesis. Further efforts are necessary to understand how to correctly modulate these mechanisms in order to reduce the pathological aspects of the altered synaptic plasticity, thus contributing to reducing the rate of neurodegeneration and/or supporting the compensatory processes that are capable of sustaining function sparing, thereby delaying symptoms, reducing the severity of functional deficits, and prolonging survival.

Until the identification of etiological factors and discovery of reliable early biomarkers or predictors, it is evident that therapies can only start after the onset of symptoms. Therefore, even the most effective therapeutic approach would only slow down the progression of disease and make the compensatory changes more efficient and long-lasting in order to overcome the functional deficits despite the permanent neuronal and synaptic loss. In this context, a better understanding of the role of early synaptic alterations, including cortical hyperexcitability and connectome alterations that can be revealed by non-invasive methods such as fMRI, would represent a future direction towards the development of methods for early presymptomatic diagnosis. Coupling early diagnosis with an effective therapy capable of stimulating synaptic plasticity would represent an ideal future scenario.

## Figures and Tables

**Table 1 ijms-24-04613-t001:** Mechanisms of synaptic plasticity and their involvement in ALS pathogenesis.

Anatomical Site	Patient or Animal Model	Type of Plastic Changes	Compensatory or Pathological?	References
Cerebral cortex	ALS patients	Reduced cortical connections	Pathological	[39]
	ALS patients	Increased neuronal excitability	Pathological	[40,41]
	ALS patients	Increased cortical connections or increased neuronal excitability	Compensatory	[42,43,44,45,46,47]
	SOD1 tg mouse	Reduced LTP	Pathological	[48]
	FUS tg mouse	Hyperexcitability; impaired synaptic plasticity	Pathological	[49,50,51]
	FUS tg Drosophila	Increased neuronal excitability	Pathological	[52]
	SOD1 tg mouse	Increased neuronal excitability	Compensatory	[53]
	TDP-43 tg mouse	Impaired synaptic plasticity, synaptic density and LTP	Pathological	[54,55,56,57]
	TDP-43 KO mouse	Impaired synaptic plasticity and LTP	Pathological	[58]
Hippocampus	ALS patients	Reduced and altered neurogenesis	Pathological	[59]
	C9orf72 KO mouse	Reduced neurogenesis	Pathological	[60]
	C9orf72 mutant patients and mice	Impaired synaptic function and plasticity	Pathological	[61,62]
	FUS tg mouse	Hyperexcitability	Pathological	[49,50,51]
	Ataxin2-mutant Drosophila	Alteration of long-term memory	Pathological	[63]
	TDP-43 tg mouse	Reduced LTP and synaptic density	Pathological	[56,57]
	TDP-43 KO mouse	Reduced LTP and synaptic density	Pathological	[64]
	TDP-43 KO rat	Increased LTP—enlarged dendritic spines	Compensatory	[65]
	SOD1 tg mouse	Excitotoxicity	Pathological	[66]
	SOD1 tg mouse	Increased synaptic plasticity and LTP	Compensatory	[66,67,68]
Spinal cord and brainstem	ALS patients	Reduced activity of inhibitory interneurons—increased MN activity	Compensatory	[69]
	ALS patients	Hyperexcitability and excitotoxicity	Pathological	[69]
	ALS patients	Reduced MN soma size and dendrites	Pathological	[70,71]
	SOD1 tg mouse	MN soma size plasticity—arborization of dendrites and proximal fibers	Compensatory	[31,33]
	SOD1 tg mouse	Hyperexcitability and excitotoxicity	Pathological	[72]
	SOD1 tg mouse	Increase of C-boutons in MNs	Compensatory	[73,74,75]
	SOD1 tg mouse	MN dendrite branching	Compensatory	[76]
	VEGF tg mouse	Impaired synaptic plasticity	Pathological	[77]
	SOD1 tg mouse	Brainstem neurogenesis	Compensatory	[78]
	Neurotoxic MN removal	Synaptic plasticity	Compensatory	[34,35,36,37,79,80,81]
	SOD1 tg rat	Phrenic long-term facilitation	Compensatory	[82,83]
NMJ	ALS patients	Neurotrophic support from muscle	Compensatory	[84]
	SOD1 tg mouse	Altered NMJ sprouting	Pathological	[85]
	Mnd mouse	Altered NMJ sprouting	Pathological	[85]
	Pmn mouse	Altered NMJ sprouting	Pathological	[85]
	SOD1 tg mouse	Detachment of synaptic terminals	Pathological	[86]
	SOD1 tg mouse	Detachment of synaptic terminals in selective NMJ subtypes	Pathological	[87,88]
	SOD1 tg mouse	Synaptic plasticity in selective NMJ subtypes	Compensatory	[87,88]
	SOD1 tg mouse	Neurotrophic support from muscle	Compensatory	[89]
	Mouse and Drosophila with mutant sodium channels	Homeostatic synaptic plasticity	Compensatory	[90]
	TDP-43 KO Drosophila	Altered synaptic function	Pathological	[91]
	C9orf72 mutant Drosophila	Homeostatic synaptic plasticity	Compensatory	[92]

Abbreviations: tg = transgenic; KO = knockout.

## Data Availability

Not applicable.

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
