# Peer review of "Synaptic Dysfunction and Plasticity in Amyotrophic Lateral Sclerosis"

_ijms, 2023, doi:10.3390/ijms24054613_

Round 1

Reviewer 1 Report

The work is a literary review on the topic of synaptic dysfunction in various parts of the nervous system in the pathogenesis of ALS. The literature data described by the author often contradict each other.

Criticism

There is a lack of certainty, a clear opinion of the author in which direction it is necessary to move in order to understand the molecular mechanisms of ALS pathogenesis. Which of the existing studies are the most promising? Which research could lead to the development of new ALS preventive drugs?

Author Response

The work is a literary review on the topic of synaptic dysfunction in various parts of the nervous system in the pathogenesis of ALS. The literature data described by the author often contradict each other.

RESPONSE: I agree with the reviewer. The literature is contradictory, and this is partially due to the relatively little number of studies in this field, which requires further efforts to clarify many aspects. However, I decided to write a review on this field, probably still relatively young, in order to underly the potential promising aspects of studying synapses in the context of ALS pathogenesis. Each one of the contradictory aspects underlined here would represent a future research project.

Criticism

There is a lack of certainty, a clear opinion of the author in which direction it is necessary to move in order to understand the molecular mechanisms of ALS pathogenesis. Which of the existing studies are the most promising? Which research could lead to the development of new ALS preventive drugs?

RESPONSE: In my opinion, the role of synapses is central in the understanding of ALS pathogenesis, and although synaptic failure is an obvious (but still requiring investigation) piece of pathogenesis, the role of synaptic plasticity as a compensatory mechanism is much more interesting, because it make the neural circuits potentially capable of resiliency. Additional comments have been included in the Conclusions (lines 482-483 and 494-499).

Reviewer 2 Report

This manuscript is a focused review of the literature on synaptic dysfunction and plasticity in ALS.  It is well-organized and discusses the topic in a balanced manner that is meant to provide food for thought about the biology of this disease.

One point that perhaps could be better emphasized is that ALS may not be a single “disease” but a syndrome of similar “diseases” that may have different etiological drivers.  Therefore animal models of this syndrome have tended to focus on the known genetic drivers or else have attempted to mimic some of the end-stage pathology using toxic interventions that may or not be found in the human disease. 

Relatively little is discussed about the upper motor neuronal element of the disease, which is variable, but in some patients is the primary deficit.

There are some instances where grammar and syntax are incorrect or awkward, making it harder to understand the point of the sentence.  Many are minor, but I am listing them here along with some specific questions to clarify what is meant.

Line 48: “C9orf72 mutations is” should be changed to ”the C9orf72 mutation is” 

Lines 59-60: “although several gene mutations can be found in all ALS cases”  What is meant by this statement?  Several  mutations in which genes?

Line 144: change “intricated” to “intricate”.

Lines 161-163: “Moreover, a decreases expression of NMDA was found in the motor cortex of presymptomatic SOD1 transgenic mice, together with reduced dendritic arborization and long-term potentiation (LTP) defects [50].”   This would read better if changed to the following:  “Moreover, a decrease in expression of NMDA was found in the motor cortex of presymptomatic SOD1 transgenic mice together with a reduction in dendritic arborization and defects in long-term potentiation (LTP) (50)”  Keeping a parallel construction increased clarity here.

Line 170:  It is unclear which presynaptic compartment is being referred to here.  Is it the afferents to the hippocampal somata or is it the axon terminals of the hippocampal neurons?  Please be more specific.

Lines 177-181:  This is a run-on sentence that should end at the semicolon in line 180.  Also the commas around “in cortical neurons” should be removed.  Begin a new sentence starting after the semicolon, “Neuronal activity ……"

Line 186:  change “inflammatory processes” to “inflammation”

Line 188: change “protein” to “protein’s”

Line 192:   It is not entirely clear what “this function” refers to.  What exactly is “partially maintained” in female mice vs. male mice?

Line 225:  “Localization” is the subject so “is” is the verb.

Lines 230-233: “For instance, alterations of synaptic functions in the hippocampus and neocortex have been reported in mice carrying a FUS mutation, as a result of dysregulated RNA trafficking, altered protein homeostasis, mitochondrial function and axonal protein synthesis, thus causing dysfunctions in brain connectivity and cognitive deficits”   Changing the wording to a more parallel form with increase the clarity of the sentence.  “For instance, alterations of synaptic functions in the hippocampus and neocortex have been reported in mice carrying a FUS mutation, as a result of dysregulation of RNA trafficking and alterations of protein homeostasis, mitochondrial function and axonal protein synthesis, thus causing dysfunctions in brain connectivity and cognition.”

Lines 234-236: “Interestingly, alterations of RNP granule assembly were found in Drosophila with ataxin-2 mutation, together with alterations of long-term memory and C9orf72- and FUS-induced neurodegeneration [75].:  This sentence is unclear.  What mutations did the flies have?  How did flies with an ataxin-2 mutation have C0orf72 and FUS induced neurodegeneration?

Lines 240-241: “causing several functional defects and degeneration due to both toxicity and loss of function [76]. “  This statement seems tautological.  Functional defects are the definition of
loss of function so they cannot be explained by restating what they are.  The sentence also seems to imply that loss of function “causes” the degeneration?  Is that what is meant?

Line 242:  What are the dementia-like deficits?  Are they behavioral or just anatomical?  If they are just anatomical, I would suggest rephrasing it to reflect that.

Line 244: change “onto” to “on”

Line 247: insert a comma after “10)”

Lines 253, 262: change “to” to “with”

Line 259: remove comma after “studies”

Line 320: insert “and” between “transport,” and “growth”

Line 321: replace the comma after “pathways” with a semicolon

Lines 327-333:  Why couldn’t this effect be due to terminal sprouting from surviving MNs if the model is knocking out only a portion of the overall population?  Presumably this toxin-induced loss of  MNs is abrupt rather than gradual as in a neurodegenerative process.

Lines 363-367:  change “associated to” to “associated with”.  Clarify which mouse model is used in citation #107, since there is a variance with what is seen in an SOD-1-model (citation #113).  How is “muscarinic activation” of Schwann cells defined?

Line 383: change “promote” to “promotes”

Lines 388-391: This sentence is unclear and needs to be rephrased. 

Line 410: Add “the” before central and change “system” to “systems”

Line 422: change “intricated” to “intricate”

Line 445: change “were also reported” to “also have been reported”

Line 451: change “demonstrate” to “demonstrates”

Line 454: omit “the” before “activity-dependent”

Lines 465-466: change “that are capable to preserve functions” to “that is capable of preserving function”

Author Response

This manuscript is a focused review of the literature on synaptic dysfunction and plasticity in ALS.  It is well-organized and discusses the topic in a balanced manner that is meant to provide food for thought about the biology of this disease.

RESPONSE: thank you so much for this appreciation.

One point that perhaps could be better emphasized is that ALS may not be a single “disease” but a syndrome of similar “diseases” that may have different etiological drivers. Therefore animal models of this syndrome have tended to focus on the known genetic drivers or else have attempted to mimic some of the end-stage pathology using toxic interventions that may or not be found in the human disease. 

RESPONSE: this is an important and interesting point. However, this review addressed a particular aspect of ALS pathogenesis, which is the involvement of synaptic function and plasticity. As noted by this reviewer (in the next comment), relatively little is known about this aspect of pathogenesis, and this lack of information would complicate the discussion in the context of ALS as a syndrome. However, it appears from the analysis of the literature, that synaptic function may be central in all aspect of the ALS syndrome. In fact, many of the available models of different species have been discussed, including those mimicking motor dysfunctions or cognitive deficits. In addition, several data from ALS and/or FTD patients are also discussed, and the involvement of synaptic function, failure and plasticity seems to be central in all studies. In the revised version of the manuscript, I tried to address this point by adding some comments in the Discussion (lines 423-428).

Relatively little is discussed about the upper motor neuronal element of the disease, which is variable, but in some patients is the primary deficit.

RESPONSE: I would like to thank the reviewer for this comment, which gives me the opportunity to clarify the method for collecting the cited literature. In particular, all relevant papers found by using the keywords “synaptic plasticity” AND “amyotrophic lateral sclerosis” have been evaluated and included in the review. A number of papers (about twenty) studying plasticity in the cerebral cortex of ALS patients or animal models were found and discussed in the section 2.1. Among these, papers showing alterations of neuronal excitability in the motor cortex are also cited, but they are probably fewer than expected, as the reviewer noted. In my view, it suggests that the research field of synaptic plasticity in the motor cortex, in the context of ALS, is open and requires more efforts. A sentence has now been added in the Section 2.1 (line 199-201) concerning these aspects.

There are some instances where grammar and syntax are incorrect or awkward, making it harder to understand the point of the sentence.  Many are minor, but I am listing them here along with some specific questions to clarify what is meant.

Line 48: “C9orf72 mutations is” should be changed to ”the C9orf72 mutation is” 

RESPONSE: done (line 48).

Lines 59-60: “although several gene mutations can be found in all ALS cases”  What is meant by this statement?  Several  mutations in which genes?

RESPONSE: the meaning of the sentence is that mutations of different genes linked to ALS can be found also in sporadic cases, not only in familial ones. I rephrased this sentence trying to make it clearer (lines 59-60).

Line 144: change “intricated” to “intricate”.

RESPONSE: done (line 145).

Lines 161-163: “Moreover, a decreases expression of NMDA was found in the motor cortex of presymptomatic SOD1 transgenic mice, together with reduced dendritic arborization and long-term potentiation (LTP) defects [50].”   This would read better if changed to the following:  “Moreover, a decrease in expression of NMDA was found in the motor cortex of presymptomatic SOD1 transgenic mice together with a reduction in dendritic arborization and defects in long-term potentiation (LTP) (50)”  Keeping a parallel construction increased clarity here.

RESPONSE: done as suggested (lines 162-164).

Line 170:  It is unclear which presynaptic compartment is being referred to here.  Is it the afferents to the hippocampal somata or is it the axon terminals of the hippocampal neurons?  Please be more specific.

RESPONSE: yes, this requires a clarification. The authors described FUS accumulation in dendrites and bodies of cortical and hippocampal neurons, and a deeper investigation demonstrated a precise localization in the presynaptic compartment of these synapses. So, it is referred to synaptic afferents to cortical and hippocampal neurons. The sentence has now been rephrased (lines 171-172).

Lines 177-181:  This is a run-on sentence that should end at the semicolon in line 180.  Also the commas around “in cortical neurons” should be removed.  Begin a new sentence starting after the semicolon, “Neuronal activity ……"

RESPONSE: done as suggested (lines 180-181).

Line 186:  change “inflammatory processes” to “inflammation”

RESPONSE: done (line 187).

Line 188: change “protein” to “protein’s”

RESPONSE: done (line 189).

Line 192:   It is not entirely clear what “this function” refers to.  What exactly is “partially maintained” in female mice vs. male mice?

RESPONSE: “this function” refers to the “dendritic spine plasticity”, which is altered in TDP-43 mutant mice, but more altered in male than in female mice. I rephrased to make the sentence clearer (lines 193-195).

Line 225:  “Localization” is the subject so “is” is the verb.

RESPONSE: mistake corrected (line 228).

Lines 230-233: “For instance, alterations of synaptic functions in the hippocampus and neocortex have been reported in mice carrying a FUS mutation, as a result of dysregulated RNA trafficking, altered protein homeostasis, mitochondrial function and axonal protein synthesis, thus causing dysfunctions in brain connectivity and cognitive deficits”   Changing the wording to a more parallel form with increase the clarity of the sentence.  “For instance, alterations of synaptic functions in the hippocampus and neocortex have been reported in mice carrying a FUS mutation, as a result of dysregulation of RNA trafficking and alterations of protein homeostasis, mitochondrial function and axonal protein synthesis, thus causing dysfunctions in brain connectivity and cognition.”

RESPONSE: done as suggested (lines 235-237).

Lines 234-236: “Interestingly, alterations of RNP granule assembly were found in Drosophila with ataxin-2 mutation, together with alterations of long-term memory and C9orf72- and FUS-induced neurodegeneration [75].:  This sentence is unclear.  What mutations did the flies have?  How did flies with an ataxin-2 mutation have C0orf72 and FUS induced neurodegeneration?

RESPONSE: yes, the sentence is misleading and it has been rephrased (237-240).

Lines 240-241: “causing several functional defects and degeneration due to both toxicity and loss of function [76]. “  This statement seems tautological.  Functional defects are the definition of
loss of function so they cannot be explained by restating what they are.  The sentence also seems to imply that loss of function “causes” the degeneration?  Is that what is meant?

RESPONSE: I agree with the reviewer that this sentence seems tautological because of some confounding terms. However, “functional defects” is referred to symptoms (ALS-typical motor and/or non-motor deficits), while “loss of function” is referred to the loss of the normal TDP-43 functions due to mutation, and “toxicity” is also referred to mutant TDP-43, being a “gain of function”. The sentence would mean that the gain of function (toxicity of aggregates) and loss of function of mutant TDP-43 may cause neuronal degeneration and the consequent functional deficits and symptoms. I rephrased the sentence to make it clearer (lines 244-245).

Line 242:  What are the dementia-like deficits?  Are they behavioral or just anatomical?  If they are just anatomical, I would suggest rephrasing it to reflect that.

RESPONSE: “dementia-like” deficits, as described by the authors of the paper [ref. no. 57], are behavioural abnormalities resembling human dementia, including perturbations of social behavior, changes of daily activity and memory loss. The sentence has been rephrased by changing the word “deficits” with “behavioural alterations” (line 247).

Line 244: change “onto” to “on”

RESPONSE: done (line 249).

Line 247: insert a comma after “10)”

RESPONSE: done (line 252).

Lines 253, 262: change “to” to “with”

RESPONSE: done (lines 258 and 267).

Line 259: remove comma after “studies”

RESPONSE: done (line 264).

Line 320: insert “and” between “transport,” and “growth”

RESPONSE: done (line 325).

Line 321: replace the comma after “pathways” with a semicolon

RESPONSE: done (line 326).

Lines 327-333:  Why couldn’t this effect be due to terminal sprouting from surviving MNs if the model is knocking out only a portion of the overall population?  Presumably this toxin-induced loss of  MNs is abrupt rather than gradual as in a neurodegenerative process.

RESPONSE: the terminal sprouting from surviving MNs is possible, but its occurrence has been indirectly hypothesized in our model, by showing a reversal of muscle atrophy [34,37]. On the other hand, plastic changes within the spinal circuitries has been directly demonstrated in the cited papers [34-37, 94-96], by showing modifications in the expression of synapsin-1, synaptophysin, glutamate receptors and other molecular correlates of synaptic plasticity, as well as by electromyography. A sentence has been added to the text to clarify this issue (lines 336-337).

Lines 363-367:  change “associated to” to “associated with”.  Clarify which mouse model is used in citation #107, since there is a variance with what is seen in an SOD-1-model (citation #113).  How is “muscarinic activation” of Schwann cells defined?

RESPONSE: please note that both papers used SOD1 mice and that the findings in refs. #107 and #113 were not necessarily in contrast. In particular, both papers demonstrated that NMJ plasticity is absent in SOD1 mice, and this was due to the expression of semaphorin (in ref. #107) or to the absence of muscarinic activation of Schwann cells (in ref. #113). Muscarinic activation is defined as functional changes of Schwann cells induced by their muscarinic receptors. Changes in the sentence have been done as suggested (lines 370-371).

Line 383: change “promote” to “promotes”

RESPONSE: done (line 390).

Lines 388-391: This sentence is unclear and needs to be rephrased. 

RESPONSE: done (lines 395-396).

Line 410: Add “the” before central and change “system” to “systems”

RESPONSE: done (line 417).

Line 422: change “intricated” to “intricate”

RESPONSE: done (line 434).

Line 445: change “were also reported” to “also have been reported”

RESPONSE: done (line 457).

Line 451: change “demonstrate” to “demonstrates”

RESPONSE: done (line 464).

Line 454: omit “the” before “activity-dependent”

RESPONSE: done (line 466).

Lines 465-466: change “that are capable to preserve functions” to “that is capable of preserving function”

RESPONSE: done (lines 477-478).